# PROBABILISTIC SPARSE VARIATIONAL MODEL BASED ON GAUSSIAN PROCESSES FOR ENERGY PARAMETER FORECASTING

**Konstantin Koshelev, Sergei Strijhak & Ilia Stulov**
Ivannikov Institute for System Programming of the Russian Academy of Sciences
109004, Moscow, Alexander Solzhenitsyn st., 25
{k.koshelev,s.strijhak}@ispras.ru

## ABSTRACT

Time series forecasting is an urgent task in engineering, climatology, economics, sociology, energy. Time series characterize a number of physical and economic processes. Forecasting methods come in short-term, medium-term, and long-term. The probabilistic Sparse Variational model based on Gaussian Process (SVGP) included a fully connected neural network model to find the mean function and the covariance function. In this reseach we focused on the possibility of using SVGP to predict energy parameters for two different examples with wind turbine and power generator. The model is trained on historical data for 2 years with a recording frequency of one hour. The prediction time for the power value varies from 24 hours to 168 hours. The prediction results included graphs for the desired value of energy parameter and the values of MAE, RMSE, $R^2$ and CRPS metrics.

## 1 INTRODUCTION

Forecasting has been an essential part of the power energy industry. The issues of studying energy consumption in large cities due to the mobility of people and urban growth are relevant.

The sources of electric power in cities can be thermal power plants, wind power plants, solar power plants, hydroelectric power plants, and others.

Time series forecasting is an urgent task in engineering, climatology, economics, sociology. Time series characterize a number of physical and economic processes. Forecasting methods come in short-term, medium-term, and long-term. Statistical models usually predict values for 1-3 points ahead, which is insufficient for many tasks. There are deterministic and probabilistic forecast models using machine learning and neural network methods in renewable energy sources.

Over the last 5 years, about 25 new wind farms with a total generation capacity of 2 GW have been built in Russia. During the operation of wind power plants, data are collected using automated control systems and processed. There is a need to develop models for forecasting power generation for several days in advance. This is necessary in connection with planning the dispatch schedule.

Wind power generation depends on atmospheric processes in nature. Wind is generated due to the difference in atmospheric pressure. There are many factors that can influence wind generation such as terrain, temperature and surface friction. Therefore, the dynamic behavior of wind is quite a complex phenomenon, resulting in high variability of wind power generation.

Wind speed is the meteorological variable that is most important for wind power generation. Therefore, it can be concluded that wind power generation is a nonlinear and unstable process. Meanwhile, the complex features of wind speed variability lead to the low predictability of wind generation.

The prediction of wind power output is part of the basic work of power grid dispatching and energy distribution. The output power prediction is mainly obtained by fitting and regressing the historical data. The medium- and long-term power prediction results exhibit large deviations due to the uncertainty of wind power generation. In order to meet the demand for accessing large-scale wind power into the electricity grid and to further improve the accuracy of short-term (hours), long-term (days)

wind power predictions, it is necessary to develop models for accurate and precise wind power prediction based on advanced algorithms for studying the output power of a wind power generation system Tsai et al. (2023).

There are open meteorological and generation data from various competitions GEFCom2014, Baidu KDD Cup 2022, Zendoo UK 2022, IEEE Power and Energy Society Working Group on Energy Forecasting and Analytics 2024.

GEFCom2014 wind power dataset includes open data from 2012 - 2013 with a data record multiplicity of one hour. The data are from Numerical Weather Prediction (NWP) model for wind speeds at 10 meters and 100 meters and normalized power. For example, the Baidu KDD Cup 2022 dataset containes 10 features for 245-day history with a data record multiplicity of 10 minutes.

A total of 2490 participants from 32 countries participated in the Baidu KDD Cup 2022 competition to predict power generation for a wind farm with 134 wind turbines. The winners of the best solutions included projects with such models as: Fusion of two very different models (Modified DLinear and Extreme Temporal Gated Network), Deep learning neural network and K-nearest neighbors model, a Gradient Boosting Tree based model, a single BERT model, and others.

Compared to currently wide-used point forecasts, the probabilistic forecasts could provide additional useful quantitative information on the uncertainty associated with wind power generation. For decision-makings in the uncertainty environment, the probabilistic forecasts are optimal inputs Zhang et al. (2014).

There are also several public datasets and reseach on energy consumption values for the private house network Wang et al. (2019). Such free datasets contain electricity consumption in kW for hundreds of households with minutes frequency and different geographical locations Ansari et al. (2024).

Researchers and practitioners have contributed thousands of papers on forecasting electricity demand and prices, and renewable generation (e.g., wind and solar power). Some articles offer a brief review of energy forecasting papers, summarize research trends, discuss importance of reproducible research and points out valuable open data sources. They make recommendations about publishing high-quality research papers and offer an outlook into the future of energy forecasting Hong et al. (2020).

For guaranteed accuracy of forecasting for a large number of points ahead, it is advisable to apply probabilistic models of forecasting in time, as they contain a confidence interval. The uncertainty of the forecasted values can be expressed using different probability measures (PDF, CDF, quantiles, intervals, variance, others).

To build a probabilistic prediction model, a Gaussian Processes (GPs) regression model can be used for the desired feature vector depending on the choice of data in the presence of noise magnitude. Additionally, Bayesian theory was used to calculate the posterior distribution, e.g., of the power plant power value, i.e., the value obtained after observations, which was the product of the a priori probability and likelihood.

The probabilistic Machine Learning Theory has been described well in the textbooks Murphy (2022; 2023). Some efficient GP methods and scientific libaries have been developed in recent years to deal with big data in combination with popular Machine Learning (ML) frameworks TensorFlow, PyTorch (Gardner et al. (2021); Wang et al. (2019); Potapczynski et al. (2023); Picheny et al. (2023)).

Some libraries are focused on methods with inducing Point Allocation for Sparse Gaussian Processes in High-Throughput Bayesian Optimization Moss et al. (2023).

Different GP model has been applied to predict wind farm power generation Wen et al. (2022) , Electrical Load Forecasting against Anomalous Events (Cao et al. (2022), approximations on different datasets with $10^4 - 10^6$ data points Wang et al. (2019)).

Analytical platforms are being developed in various domains for decision making under uncertainty using Bayesian statistics and GP regression models. Some examples of using machine learning techniques for demand forecasting, relying on the transformer architecture, in the online industry are presented in the articles Kunz et al. (2023); Ansari et al. (2024).

In our work, we would like to focus on the possibility of using GPs to predict energy parameters for two different examples. The article highlights aspects of mathematical theory in the use of GPs.

## 2 MATHEMATICAL MODEL

Deep neural networks are a family of flexible approximators of functions of the form $f(x, \theta)$, where the dimensionality $\theta$ is fixed and does not depend on the size $N$ of the training data set.

However, such parametric models may be prone to overtraining at small $N$ and undertraining at large $N$ due to the fixed capacity. We will use nonparametric models to build models whose capacity automatically adapts to the amount of data.

There are many different approaches to building nonparametric models for classification and regression. We will consider a Bayesian approach in which the uncertainty of the mapping of inputs to outputs $f$ is represented using an a priori distribution of features and then updating it based on the collected data.

Commonly used deterministic Power Foresting models with Numerical Weather Predictions (NWP) as inputs can be mathematically described as:

$$\hat{y} = g(x; w),$$

where $g$ denotes the forecasting model, $x$ denotes NWPs, $\hat{y}$ denotes output, i.e., the estimated wind power generation, and $w$ denotes its parameters.

Generally, the forecasting model can be expressed as

$$g(x; w) = \phi(x)^T w,$$

where $\phi(x)^T$ denotes a basis function that transforms $x$ into a vector of feature space.

The uncertainty is aroused by an aleatory error, denoted by $\epsilon$ to characterize the difference between the latent value $f$ (that is specified in a power curve of generator) and the observation value $y$:

$$\epsilon = y - f,$$

We have

$$y = g(x; w) + \epsilon.$$

that $\epsilon$ is normally distributed with zero mean and variance $\sigma^2$ obs. The epistemic uncertainty is caused by the limited representation capability of NWPs.

It accounts for the uncertainty in modeling procedure from NWPs to wind power generation. That is, given a set of inputs $[x_1, \ldots, x_k]$ with equivalent value, the corresponding latent values $[f_1, \ldots, f_k]$ are different.

We use a Gaussian process (GP) to represent the a priori distribution $p(f)$, followed by a formula from Bayes' theorem to derive the posterior distribution $p(f|D)$, which is another GP Rasmussen & Williams (2006).

Recall that a Gaussian random vector of length $N$, $f = [f_1, \ldots, f_N]$, is defined by its mean $\mu = E[f]$ and covariance matrix $\Sigma = Cov[f]$. Now consider the function $f : X \to R$ computed on the set of inputs $X = \{x_n \in X\}$.

We denote by $fx = [f(x_1), \ldots, f(x_n)]$ the set of unknown values of the function at these points.

If $fx$ has a Gaussian distribution for any set $N \geq 1$ points, then $f : X \to R$ is said to be a Gaussian process. Such a process is defined by its mean function $m(x) \in R$ and covariance function $K(x, x) \geq 0$, which can be any positive definite kernel. For example, a radial-basis kernel can be used.

Gaussian Processes (GPs) are non-parametric machine learning models that place a distribution over functions f $\sim$ GP.

We denote the corresponding Gaussian process:

$$f(X) \sim GP\ \left(m(X),\ K\left(X,\ X^{'}\right)\ \right)$$

The main parameters for GP process are:

$m(X) =\ E[f(X)]$ – mean function

$K\left(X,\ X^{'}\right) =\ E\left[(f(X) -\ m(X))\left(f\left(X^{'}\right) -\ m\left(X^{'}\right)\right)^{\top}\right]$ - covariance function

$$\begin{bmatrix} Y \\ f_* \end{bmatrix} \sim\ \mathcal{N}\left(0, \begin{bmatrix} K(X,X) + \sigma_n^2 I & K\left(X,X_*\right) \\ K\left(X_*,\ X\right) & K\left(X_*,\ X_*\right) \end{bmatrix}\right)$$

$K(X, X)$ represents the covariance matrix on inputs of training set

$K\left(X, X_*\right)$ represents the vector of covariance between the test point and the training inputs

$K\left(X_*,\ X_*\right)$ is the covariance of the test points

$\sigma_n^2 I$ is the variance multiplying an identity matrix with appropriate dimensions

$$f_*|X,\ Y,\ X_*\ \sim\ \mathcal{N}\left(\mu_*,\ \Sigma_*\right)$$

$\mu_* = K\left(X_*,\ X\right)\left[K(X,\ X) + \sigma_n^2 I\right]^{-1} Y$

$\Sigma_* = K\left(X_*,\ X_*\right) - K\left(X_*,\ X\right)\left[K(X,X) + \sigma_n^2 I\right]^{-1} K\left(X,\ X_*\right)$

$$Y = f(X) + \epsilon,\ \epsilon\ \sim\ \mathcal{N}\left(0,\ \sigma_n^2\right)$$

In our study, we employed the Radial Basis Function (RBF) kernel as the covariance function for the Gaussian Process model. The RBF kernel, also known as the squared exponential kernel, is widely used in machine learning due to its ability to model smooth and continuous functions. The RBF kernel assumes infinitely differentiable functions, making it suitable for modeling processes with strong smoothness assumptions.

This kernel is defined as:

$$\mathcal{K}(x, x'; \ell) = \exp\left(-\frac{\|x - x'\|^2}{2\ell^2}\right)$$

where:

- $\ell$ is the length scale parameter, which controls how far the influence of a single training point reaches. A larger $\ell$ results in smoother functions, while a smaller $\ell$ allows more flexibility in capturing local variations.
- $\|x - x'\|$ represents the Euclidean distance between input points $x$ and $x'$. This ensures that points closer in the input space have higher correlation, while points further apart have a lower correlation.

The RBF kernel provides a strong smoothness assumption, making it effective for modeling energy forecasting problems where continuity and differentiability are critical. However, unlike the Matérn kernel, it may be more sensitive to overfitting in datasets with abrupt changes or discontinuities. Despite this, the RBF kernel remains a preferred choice due to its stability and widespread applicability in Gaussian Process modeling.

The quadratic exponential kernel hyperparameters with variance scale parameter $\sigma_f$:

$$K\left(x,\ x^{'}\right) = \sigma_f^2 \exp\left(-\frac{\left\|x - x^{'}\right\|^2}{2l^2}\right)$$

Most kernels include hyperparameters $l$, such as the lengthscale, which must be fit to the training data. In regression, $l$ are typically learned by maximizing the GP's log marginal likelihood with gradient descent. A typical GP has very few hyperparameters to optimize and therefore requires fewer iterations of training than most parametric models.

## 2.1 SVGP MODEL

The probabilistic sparse variational model based on Gaussian Process (SVGP) is used to reduce the dimensionality of data and reduce the required computational resources. It includes a fully connected neural network model to find the mean function and the covariance function. The covariance function has a quadratic exponential kernel.

Both the variance of time series, the covariance matrix should be optimized during the training. These parameters and some others are updated by maximizing the marginal likelihood function. Since the training time complexity of GP is $\mathcal{O}(N^3)$, it is not suitable for applications with a large number of training data. In this context, the SVGP method was developed.

By introducing $M$ inducing points to approximate the original GP, the complexity can be reduced to $\mathcal{O}(NM^2)$. Variational inference is introduced to reduce the computational burden.

It aims to look for an approximated posterior through the minimization of Kullback Leibler divergence between the true posterior $p$ and the variational posterior $q$.

The minimization problem is equivalent to the maximization of the lower bound. This allowed to reduce the dimensionality of the problem and significantly save RAM resources of our computer.

The SVGP model is described in details in references (Hensman et al. (2013); Murphy (2023))

SVGP model has been applied to predict wind turbine's power generation and energy consumption in one industrial district of a large Russian city.

The prediction results included graphs for the desired value of energy parameter and the values of MAE, RMSE, $R^2$ and CRPS metrics.

## 2.2 FEATURES OF THE MATHEMATICAL MODEL

For convenience, the power forecast probability distribution was normalized using a logit-normal transformation and varied over the range [0-1].

$$\xi = l(y) = \ln\left(\frac{y}{1-y}\right),$$

Logit-normal transformation    $y \in (0,1)$

$$p(y) = \frac{1}{\sqrt{2\pi\sigma^2}} \cdot \frac{1}{y(1-y)} \exp\left[-\frac{1}{2}\left(\frac{\ln\left(\frac{y}{1-y}\right) - \mu}{\sigma}\right)^2\right].$$

Application of transformation to GP probability

$$\begin{cases} f \sim \mathcal{GP} \\ l(y) = f + \epsilon; \quad \epsilon \sim \mathcal{N}(0, \sigma_{obs}^2 I) \end{cases}$$

$$p(\xi^*|x^*) \sim \mathcal{N}(\xi^*|\mu_f(x^*), \sigma_f^2(x^*) + \sigma_{obs}^2).$$

$$[q_{\beta/2}(x^*), q_{1-\beta/2}(x^*)],$$

$q_\alpha(x^*)$ represents the $\alpha$-th quantile of $p(y^*|x^*)$.

$$p(f|X) = \mathcal{N}(f|\mu, \mathbf{K}),$$

Under the assumption of Gaussian noise

$$p(\xi|f) = \mathcal{N}(\xi|f, \sigma_{obs}^2 I).$$

Gaussian process contains a set of random variables, any finite number of which has a joint Gaussian distribution.

### Neural network model for mean function prediction

A fully connected neural network model can be used for mean function prediction and can be described as:

$$h_k = T^{(k)}(h_{k-1}) = W_k h_{k-1} + b_k,$$

We have used the ReLU activation function:

$$\sigma(h_k)_i = \max(h_{k,i}, 0).$$

The final function is a composition of affine functions and activation functions.

$$m(x) = (T^{(L)} \circ \sigma \circ T^{(L-1)} \circ \cdots \circ \sigma \circ T^{(1)})(x),$$

### Approximated Sparse GP

The basic idea behind SVGP is that a function $f$ is defined on a finite set of possible inputs, which we partition into three subsets: the training set, the set of auxiliary points $M$, and the set of all other points that we can treat as the test set.

Methods with auxiliary points approximate the posterior distribution $p(f|y)$ c using a variational approach.

The auxiliary points $M$ and auxiliary values $fm$ are variational parameters, not model parameters, which eliminates the risk of overfitting. It can be shown that as the number of auxiliary points $M$ increases, the quality of the posterior observation can be improved. The quality of the posterior observation increases t eventually turns out to be the same as for exact inference.

We performed Complexity Reduction: $\mathcal{O}(N^3)$ to $\mathcal{O}(M^2 N)$.

M – model points $(M \ll N)$

$\mathbf{Z} = [\mathbf{z}_1, \ldots, \mathbf{z}_M]$ – set of model points

$$p(u|Z) = \mathcal{N}(u|\mu_M, K_{MM}).$$

where: $K_{MM}$ – Covariance matrix

u = g(Z) – Vector

$$\begin{pmatrix} u \\ f \end{pmatrix} \sim \mathcal{N} \left( \begin{pmatrix} \mu_M \\ \mu \end{pmatrix}, \begin{pmatrix} K_{MM} & K_{MN} \\ K_{NM} & K \end{pmatrix} \right).$$

$$p(f|u, X, Z) = \mathcal{N}(f|\mu^*, \Sigma^*),$$

$$\mu^* = \mu + K_{NM} K_{MM}^{-1}(u - \mu_M),$$

$$\Sigma^* = K - K_{NM}K_{MM}^{-1}K_{MN}.$$

**Inducing Points and Variational Inference in SVGP**

Sparse Variational Gaussian Processes (SVGP) use inducing points to approximate the full GP posterior, significantly reducing computational complexity. The following formulas illustrate the core mathematical framework underlying SVGPs, emphasizing the role of inducing points.

**Inducing Point Optimization**:

The process of selecting optimal inducing points is based on minimizing the Kullback-Leibler (KL) divergence between the approximate posterior and the true posterior. The optimization criterion is given by:

$$Z = \arg\min \text{KL}(q(u)||p(u|X, Z)) \tag{1}$$

where:

- $Z$ – the set of inducing points to be optimized.
- $q(u)$ – the variational distribution over the inducing variables.
- $p(u|X, Z)$ – the true posterior distribution of the inducing variables given the training data $X$ and inducing points $Z$.
- $\text{KL}(\cdot||\cdot)$ – the Kullback-Leibler (KL) divergence, which measures the difference between two probability distributions.

This criterion ensures that the selected inducing points $Z$ lead to an approximate posterior $q(u)$ that is as close as possible to the true posterior $p(u|X, Z)$. By minimizing the KL divergence, we improve the quality of the sparse Gaussian process approximation while maintaining computational efficiency.

**Joint Density**

The joint density of all variables involved in the SVGP model is given by:

$$p(X, Z, \xi, u, f) = p(X, Z)p(f|X, u)p(u|Z)p(\xi|f)$$

Here:

- $X$ - Input data points
- $Z$ - Set of inducing points
- $\xi$ - Observations
- $u$ - Latent values at the inducing points
- $f$ - Latent function values

This formula specifies the generative process of the SVGP model, establishing dependencies among all random variables. The set of inducing points $Z$ is crucial as they influence the entire model, enabling scalable inference through variational approximation.

**Conditional Density**

The conditional dependencies among observations, set of inducing points, and latent function values are expressed as:

$$p(\xi, u, f|X, Z) = p(f|X, u)p(u|Z)p(\xi|f)$$

This equation shows the probabilistic relationship between the observations and set of inducing points, forming the basis for variational approximation. It enables scalable inference by reducing the dependency on the full dataset.

**Variational Evidence Lower Bound**

It presents the formulation of the Variational Lower Bound used in SVGP. This approach allows for efficient and scalable inference by introducing inducing points, which serve as a set of representative data points for approximating the posterior distribution.

The ELBO in the context of SVGP is represented as follows:

$$\log p(\xi, u | X, Z) = \log \int p(f|X,u) p(u|Z) p(\xi|f) df \geq \mathbb{E}_{p(f|X,u)} \log p(\xi|f) + \log p(u|Z) = \mathcal{L}_1$$

This equation represents the Evidence Lower Bound (ELBO) used for approximating the marginal likelihood in SVGP models. The left-hand side defines the logarithm of the joint marginal likelihood of the observations $\xi$ and the inducing points $u$ given the inputs $X$ and $Z$. The right-hand side introduces a variational approximation by applying Jensen's inequality, resulting in a lower bound $\mathcal{L}_1$.

This approximation separates into two components: the expected log-likelihood term and the log-prior term over the inducing points. By leveraging inducing points, this formulation maintains scalability while preserving the probabilistic structure of Gaussian Processes, effectively reducing the computational complexity from $\mathcal{O}(N^3)$ to $\mathcal{O}(NM^2)$, where $N$ is the number of observations and $M$ is the number of inducing points.

To factorize across the data, the lower bound is calculated as follows:

$$\mathcal{L}_1 = \sum_{i=1}^{N} \log \mathcal{N}(\xi_i | k_i^T K_{MM}^{-1} u, \sigma_{obs}^2) - \frac{1}{2\sigma_{obs}^2} \mathrm{Tr}(\Sigma^*) + \log p(u|Z)$$

Equation decomposes the ELBO into three components:

- The first term calculates the log-likelihood of the observations given the inducing points, using the multivariate normal distribution.

- The second term regularizes the model by penalizing the trace of the posterior covariance, effectively preventing overfitting.

- The third term represents the prior distribution over the inducing points, ensuring a coherent probabilistic model.

These formulations are fundamental to the SVGP approach, as they allow for efficient computation while maintaining the flexibility of a full Gaussian Process.

These formulas are integral to the SVGP model's variational inference framework, specifically highlighting the role of inducing points in reducing computational complexity and maintaining model accuracy. By including them, the paper provides a solid theoretical foundation for SVGP while emphasizing the computational advantages brought by inducing points.

The parameters and hyperparameters of the proposed model are described in Table 1.

Table 1: Parameters and hyperparameters of the proposed model.

| Category | Description |
|---|---|
| Parameters | weights and biases in mean function $\{W_k, b_k\}$, noise $\sigma_{obs}$, kernel hyperparameters $\theta$, set of inducing points $Z$, mean and variance of variational distribution $m$ and $S$ |
| Hyperparameters | number of inducing points, layers and units in the mean function |

## 3 Definition of the problem

Forecasting energy consumption and production is a critical task in power systems management, ensuring optimal resource allocation, grid stability, and economic efficiency. Traditional forecasting models often rely on deterministic methods, which may not effectively capture the inherent uncertainty in energy systems caused by fluctuating demand, weather conditions, and operational constraints.

In this work, we focus on probabilistic forecasting of energy parameters using SVGP model. Unlike classical regression approaches, SVGP provides not only point predictions but also uncertainty estimates, which are essential for robust decision-making in power grid management.

The primary challenges in energy forecasting include:

- **High variability in energy data:** Energy consumption and production are influenced by multiple external factors such as weather conditions, time of day, and grid load fluctuations.
- **Scalability:** GPs typically have high computational complexity, making them infeasible for large datasets. The use of SVGP mitigates this issue by introducing a set of inducing points that approximate the full GP.
- **Model uncertainty:** Many forecasting models provide deterministic outputs without considering uncertainty, which can be problematic for decision-making under uncertain conditions.
- **Feature selection and data representation:** Identifying the most relevant features (e.g., weather parameters, historical energy data, cyclical patterns) significantly impacts model performance.

The main problem is related to the ability to forecast the electricity parameters for a wind turbine and an electric generator for a period of 1 to 7 days ahead. Thus, the model should be able to predict parameters 168 points ahead, including the mean and confidence interval.

We used historical data on power generation for 2 years with a parameter record multiplicity of 1 hour. We also aim to evaluate the capability of the SVGP algorithm, taking into account the influence of the number of inducing points on the prediction accuracy and the computer's time spent on the learning process.

To address these challenges, we constructed a dataset that integrates historical energy generation data with meteorological parameters, aiming to improve forecast accuracy. The dataset included measurements from multiple power generators and wind turbine over an extended period, incorporating features such as temperature, humidity, precipitation, atmospheric pressure, cloud cover, and time-based indicators (year, month, day, hour).

This study aims to evaluate the performance of SVGP for energy forecasting by:

- Comparing different kernel functions to determine their impact on prediction accuracy.
- Analyzing the effect of the number of inducing points on forecast performance.
- Investigating whether feature engineering techniques such as encoding cyclic time features (e.g., sin/cos transformations for hour-of-day) can enhance predictive capability.

## 3.1 DATASET1

To develop and evaluate the forecasting model, data from multiple sources were collected, including SCADA's system and "Open-Meteo source" (`https://open-meteo.com/`).

The SCADA's system dataset provided measurements from two 110kV power generators, specifically with internal numbers 1, 12, covering both active and reactive power (inflow and outflow) for each generator. The data spanned from January 1, 2023, at 00:00:00 to December 31, 2024, at 00:00:00, recorded at different time intervals. Additionally, meteorological data were sourced from "Open-Meteo source" for the same period, including:

- **Temperature (°C)**
- **Humidity (%)**
- **Precipitation (mm)**
- **Pressure (hPa)**
- **Cloud cover (%)**

However, "Open-Meteo source" provides data only at hourly intervals, making it necessary to resample the SCADA's system data to a one-hour frequency to maintain consistency across both datasets.

Regarding time series encoding, it was decided to split the timestamp column into separate features: **year, month, day, and hour**. This structure provides the model with explicit temporal information, avoiding potential issues associated with datetime encoding. The temperature at the geographical location of power generator is presented on the figure 1. The temperature value has a distinctly seasonal behavioral pattern during 2 years.

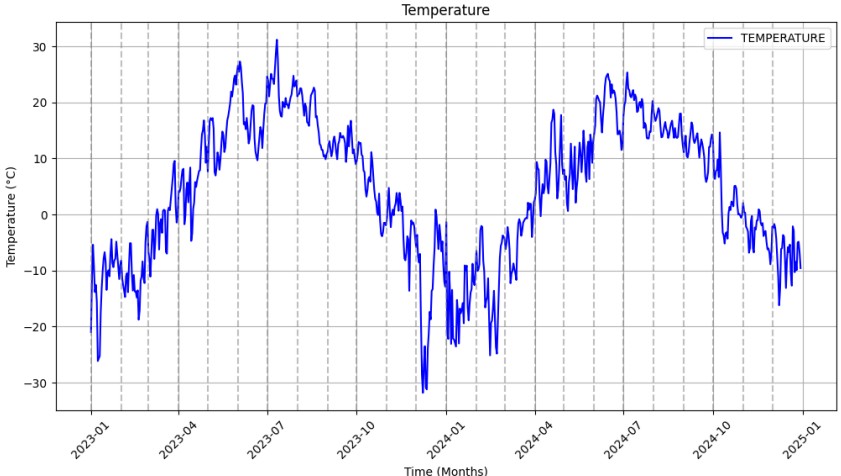

Figure 1: Temperature at the location of power plant

The value of power for generator is presented on the figure 2.

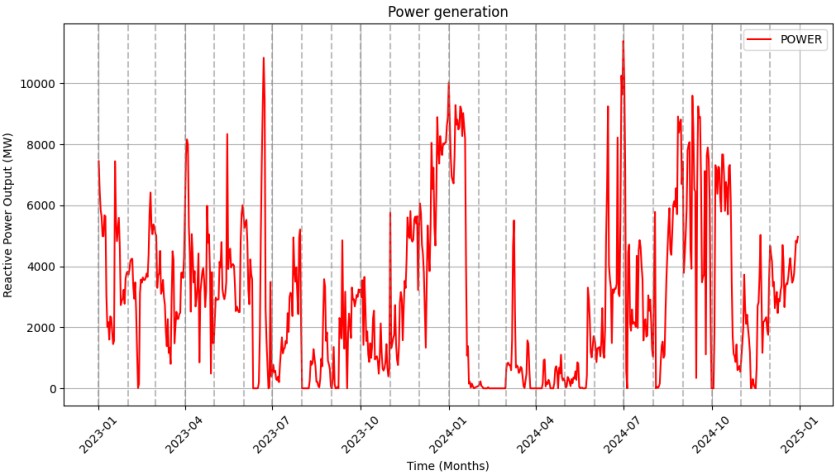

Figure 2: Power value of the power generator

As a result, a structured dataset was created, with its schema presented in Table 2.

## 3.2 DATASET2

The SCADA's system dataset provides measurements from horizontal wind turbine with 3 blades and with maximum value of power P=4.2 MWt. The wind farm with 24 wind turbines is located in a flat area, so relief and surface roughness are not considered as features. The SCADA's system allows the collection of various meteorological, mechanical and electrical parameters. The data, which were selected as velocity of wind, temperature, wind direction, power generation spanned from January 1, 2022, at 00:00:00 to December 31, 2023, at 00:00:00, have been recorded at time intervals with one hour. When analyzing the temperature values, the following was found an annual

Table 2: Feature schema of the constructed dataset

| Feature | Description |
|---|---|
| GEN1_ACTIVE_OUT | Active power output (Generator 1) |
| GEN1_ACTIVE_IN | Active power input (Generator 1) |
| GEN1_REACTIVE_OUT | Reactive power output (Generator 1) |
| GEN1_REACTIVE_IN | Reactive power input (Generator 1) |
| GEN2_ACTIVE_OUT | Active power output (Generator 2) |
| GEN2_ACTIVE_IN | Active power input (Generator 2) |
| GEN2_REACTIVE_OUT | Reactive power output (Generator 2) |
| GEN2_REACTIVE_IN | Reactive power input (Generator 2) |
| TEMPERATURE | Ambient temperature (°C) |
| HUMIDITY | Relative humidity (%) |
| PRECIPITATION | Precipitation (mm) |
| PRESSURE | Atmospheric pressure (hPa) |
| CLOUD_COVER | Cloud cover (%) |
| YEAR | Year of measurement |
| MONTH | Month of measurement |
| DAY | Day of measurement |
| HOUR | Hour of measurement |

trend. The data were further processed to eliminate outliers and the presence of zeros. The total number of entries was 17250. Data with zero values were not used in training the SVGP model.

## 4    RESULTS

The results of this work will contribute to improving probabilistic forecasting models in the energy sector, providing more reliable estimates with quantified uncertainty.

### 4.1    MODEL SELECTION AND CONFIGURATION

To enhance the accuracy and stability of the forecasting model, we employed the Radial Basis Function (RBF) Kernel instead of the Matérn Kernel. While the Matérn Kernel provides flexibility in modeling non-smooth variations, it may lead to overfitting in certain cases.

- The RBF Kernel was chosen due to its ability to model smooth functions while maintaining generalization, reducing the risk of overfitting.
- We analyzed the impact of the number of inducing points on model accuracy, demonstrating how performance improves as the number increases.

These approaches resulted in a more stable and efficient model, improving predictive performance while maintaining computational feasibility.

The results of training are presented on the figure 3.

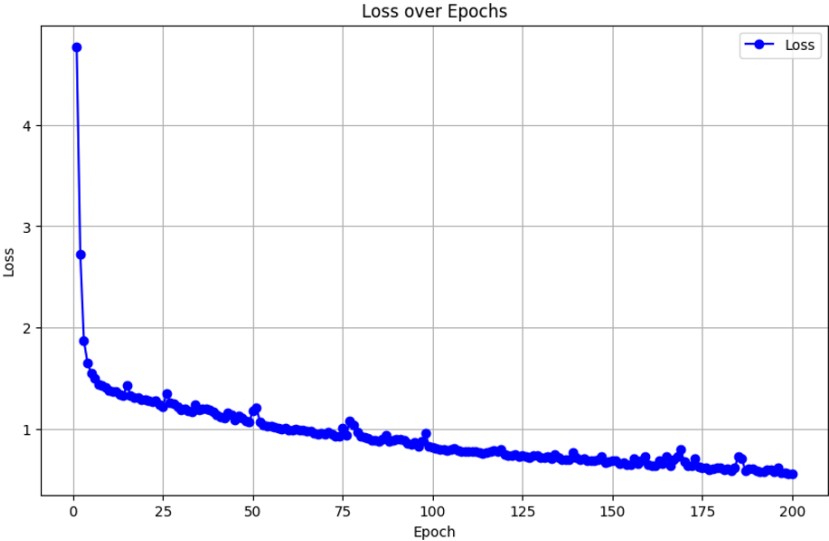

Figure 3: Traning results.

### 4.2    FORECASTING AND MODEL EVALUATION

The model was tested on two different forecasting horizons:

- 24-hour prediction
- 168-hour (7-day) prediction

For both cases, 9 features were used, and 5% of the dataset was assigned as inducing points number for training process. The model performance was evaluated using the following metrics:

### 4.2.1    24-HOUR FORECAST (9 FEATURES, 5% INDUCING POINTS)

The 24-hour forecasting results demonstrate that the model accurately captures short-term variations in energy output. The confidence interval is relatively narrow, indicating a high level of certainty in predictions. The performance metrics for this forecast are presented in Table 3.

Table 3: Evaluation metrics for the 24-hour forecast

| Metric | Definition | Value |
|--------|------------|-------|
| CRPS | Continuous Ranked Probability Score | 0.0234 |
| MAE | Dimensionless Mean Absolute Error | 0.00518 |
| RMSE | Dimensionless Root Mean Square Error | 0.00643 |
| $R^2$ | Coefficient of Determination | 0.99625 |

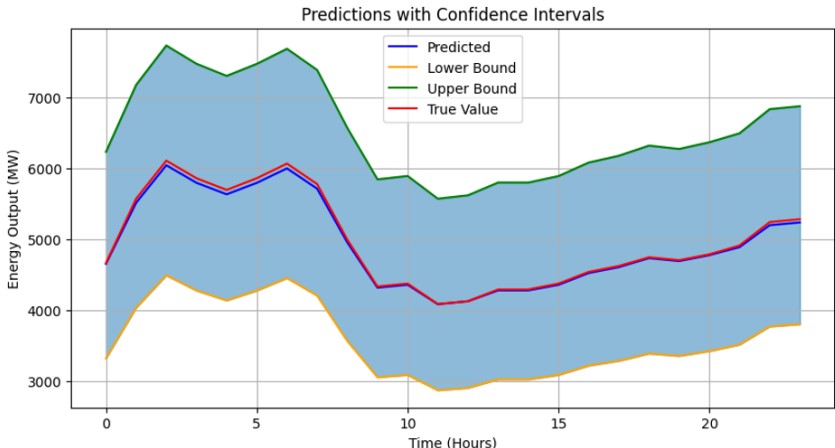

Figure 4: 24-hour forecast results. The red line represents the actual values, while the blue line represents the predicted values. The shaded region indicates the confidence interval of the prediction.

### 4.2.2 168-HOUR FORECAST (9 FEATURES, 5% INDUCING POINTS)

For the 168-hour forecast, the model successfully captures periodic trends in energy output, with increased uncertainty at longer time horizons. The confidence interval expands as the prediction period lengthens, reflecting the challenge of long-term forecasting. The performance metrics for this forecast are presented in Table 4.

Table 4: Evaluation metrics for the 168-hour forecast

| Metric | Definition | Value |
|--------|------------|-------|
| CRPS | Continuous Ranked Probability Score | 0.0229 |
| MAE | Dimensionless Mean Absolute Error | 0.0457 |
| RMSE | Dimensionless Root Mean Square Error | 0.0471 |
| $R^2$ | Coefficient of Determination | 0.9421 |

These results confirm that the model effectively captures energy consumption patterns, demonstrating low error rates across different time horizons. However, a more detailed analysis is needed to explore possible improvements in long-term forecasting accuracy.

### 4.3 UNCERTAINTY ANALYSIS AND POTENTIAL IMPROVEMENTS

The figures 4, 5 illustrate both the model's predictions and the uncertainty distribution over the forecasted periods. Based on the evaluation metrics, the model demonstrates high accuracy in predicting energy output for both the 24-hour and 168-hour horizons. However, the uncertainty bands indicate that the model encounters some difficulties in precisely estimating the range of possible values.

To address these challenges, several areas require further investigation:

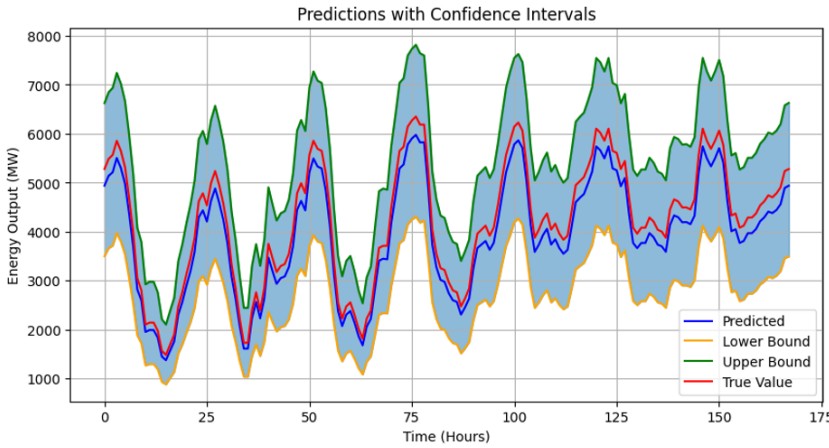

Figure 5: 168-hour forecast results. The long-term prediction shows a periodic pattern in energy consumption, with increased variance in confidence intervals over time.

1. **Number of Inducing Points** If the number of inducing points is too low, the SVGP model may struggle to accurately approximate the distribution boundaries, leading to overly narrow or excessively wide uncertainty intervals.

2. **Feature Relevance and Data Representation** The dataset's explanatory power might be insufficient, either due to a lack of relevant data or suboptimal selection of inducing points. Some features may not exhibit strong dependencies on others, contributing to uncertainty.

3. **Incorporating Sinusoidal Transformations for Periodicity** Instead of using raw time-series data, applying sinusoidal transformations such as $\sin(\text{hour})$ and $\cos(\text{hour})$ could better capture periodic patterns in energy consumption.

By refining these aspects, the predictive accuracy and reliability of the model can be further improved, particularly in reducing uncertainty intervals and enhancing long-term forecasts.

## 4.4 PREDICTIONS OF ELECTRICITY GENERATION FROM A WIND TURBINE IN A WIND FARM

In addition to forecasting energy output for power generators, an analysis was conducted to assess the applicability of the developed probabilistic sparse variational Gaussian process model to wind farms. Wind energy presents unique forecasting challenges due to its inherent variability, dependence on meteorological conditions, and the non-stationary nature of wind speed patterns.

Figures 6 and 7 illustrate the probabilistic forecast results for wind energy output over 24-hour and 48-hour time horizons, respectively. The predictions follow the expected wind power fluctuations, capturing periodic variations. However, the confidence intervals highlight the increased uncertainty associated with longer forecasting periods, emphasizing the need for additional feature engineering and potential kernel adjustments to improve accuracy.

The prediction results for the power value P of the CRPS, MAE, RMSE, $R^2$ metrics for a single wind turbine and 24 hours forecast are shown in Table 5. The maximum error value did not exceed 6 %.

Table 5: Evaluation metric for a single wind turbine and for the 24-hour forecast

| Metric | Definition | Value |
|--------|------------|-------|
| CRPS | Continuous Ranked Probability Score | 0.034 |
| MAE | Dimensionless Mean Absolute Error | 0.037 |
| RMSE | Dimensionless Root Mean Square Error | 0.041 |
| $R^2$ | Coefficient of Determination | 0.899 |

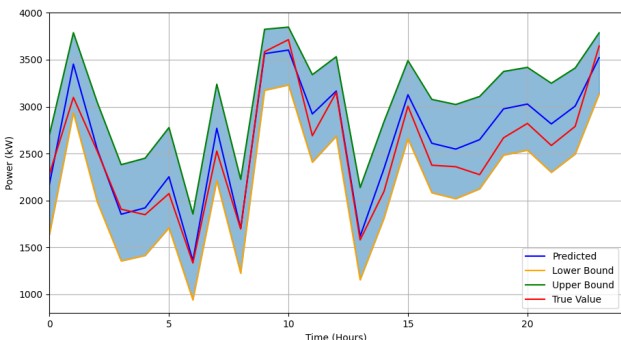

Figure 6: Probabilistic forecast of wind energy output for a 24-hour period. The shaded region represents the confidence interval.

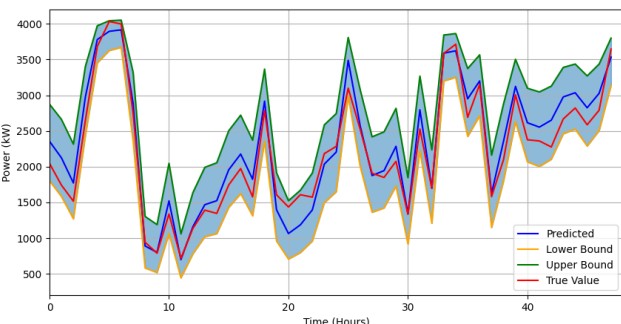

Figure 7: Probabilistic forecast of wind energy output for a 48-hour period. The uncertainty increases over a longer time horizon.

The prediction results for the power value P of the CRPS, MAE, RMSE, $R^2$ metrics for a single wind turbine and 48 hours forecast are shown in Table 6. The maximum error value did not exceed 6 %.

Table 6: Evaluation metric for a single wind turbine and for the 48-hour forecast

| Metric | Definition | Value |
|--------|-----------|-------|
| CRPS | Continuous Ranked Probability Score | 0.033 |
| MAE | Dimensionless Mean Absolute Error | 0.037 |
| RMSE | Dimensionless Root Mean Square Error | 0.045 |
| $R^2$ | Coefficient of Determination | 0.949 |

This experiment demonstrates the flexibility of the proposed SVGP-based forecasting approach in modeling different types of energy sources. The results indicate that while short-term predictions are reliable, longer-term forecasting for wind farms requires further refinement due to increased uncertainty. Future work will explore enhanced input feature selection and kernel tuning to improve performance in wind energy applications.

Our software stack includes PyTorch, gpytorch, scikit-learn, tqdm, properscoring, numpy, pandas, matplotlib, CUDA.

## 4.5 INVESTIGATING THE EFFECT OF INDUCING POINTS SELECTION ON THE MAE METRIC

We performed an additional study on selecting the number of points to predict the value of power generation 168 hours ahead for a generator. The results of the study are summarized in Table 7. The highest accuracy is obtained on the variant with 10% of points, but the training time increased 3 times in comparison with 5% of inducing points.

Table 7: Investigation of the number of inducing points on prediction performance through the MAE metric.

| Inducing Points (% of dataset) | Training Time (s) | MAE |
|:---:|:---:|:---:|
| 867 (5%) | 449 | 0.0457 |
| 1239 (7%) | 704 | 0.0250 |
| 1735 (10%) | 1346 | 0.0087 |

Table 7 presents a comparative analysis where the number of inducing points was varied to examine its impact on model accuracy in the context of electricity forecasting for a thermal power plant. For comparison, two key metrics were considered: training time and Mean Absolute Error (MAE). The results demonstrate that increasing the number of inducing points leads to a decrease in MAE, improving the model's accuracy. However, this improvement comes at the cost of significantly increased training time.

## 5 DISCUSSION

To find the model parameters, the problem of finding the maximum of the target likelihood function was solved.

The neural network model included an input layer [1,512], 2 hidden layers [512,512], one output layer [512,1]. The activation function was given by ReLu. The model was trained on a portion of historical data. Then, test data were fed to the input of the model and prediction of P value for one wind turbine was made.

Based on the results of training and prediction, the mean value of the power function and the variance value were calculated. The model was trained using Adam optimizer, lr=1e-3, number of epochs was 200. The value of M the inducing points varied in the range of 867-1735.

We have received first results for two different problems. To improve the prediction results, it is necessary to expand the number of features and to study the influence of hyperparameters on the values of the obtained metrics. The values of various metrics MAE, RMS, CRPS, $R^2$ have been obtained. The assignment of quantitative inducing points for SVGP model also requires further study.

## 6 CONCLUSION

We studied a scientif topic regarding GP models and and the peculiarities of working with big data. We performed predictions of energy performance from one to seven days using the SVGP model.

For the case with wind turbine, it is planned to develop the model taking into account new features related to the control of the wind turbine itself (blade and nacelle rotation angles), as well as for the case of a group of wind turbines that constitute a single electricity staging group.

In the future, to solve the problem with prediction of energy consumption in the city, it is planned to take into account data on the mobility of people. For example, data on the change in the number of people moving between districts due to the construction of large facilities for visiting work.

### AUTHOR CONTRIBUTIONS

The contributions of each author to this research are as follows:

Conceptualization: Author 1 and Author 3; Methodology: Author 3 and Author 1; Validation: Author 3 and Author 2; Formal Analysis: Author 2; Investigation: Author 2; Resources: Author 3; Data Curation: Author 3; Writing—Original Draft Preparation: Author 1, Author 2, and Author 3; Writing—Review and Editing: Author 3; Visualization: Author 2; Supervision: Author 2; Project Administration: Author 1; Funding Acquisition: Author 2.

All authors have read and agreed to the published version of the manuscript.

DATA AVAILIBILITY

Datasets and source code of the program on Python can be obtained from the authors upon additional request.

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
