# OpenReview forum: "Probabilistic sparse variational model based on Gaussian processes for energy parameter forecasting"
_mathai.club/MathAI/2025/Conference — MathAI 2025 Oral_

### Official Review · Reviewer_wc4U · 2025-02-24
**The article is good but there are points that require correction and addition**

**Rating:** 6
**Confidence:** 3

**Review:**

Essence:

1. There is a deficiency in the description of existing methodologies. What advantages and novel contributions does the energy forecasting approach proposed by the authors offer? In lines 077-079, it is stated that the issue was addressed using the method introduced by the authors. How does proposed approach compare to the method referenced in those lines? Additionally, there is no discussion of the current methods used for predicting energy parameters;

2. The authors utilize a fully connected neural network for time series prediction. While the authors elaborate on the processing of time series (lines 452-455) and emphasize the significance of addressing the temporal aspect of the data (lines 026-030, 421-425), they employ a fully connected layers with two hidden layers as their neural network (line 647). There is a lack of justification for the choice of a neural network that disregards the time component. Why were recurrent neural networks or transformers, which are typically employed for time series challenges, not utilized?

3. There is insufficient justification for the selection of these specific features for training (lines 441-450). In lines 036-037, the authors describe that the energy prediction is influenced by the terrain and surface friction, why these features are not used in training?

4. There is no rationale provided for the choice of metrics used to evaluate the accuracy of the problem-solving approach (lines 525-532). It would also be beneficial to include the value of the coefficient of determination;

5. A graph depicting the loss function during training is absent. Consequently, it is impossible to ascertain whether overfitting has occurred;

6. An explanation regarding the normalization of features is needed. The tables 3-4 present metric values with an accuracy of four decimal places and the order of values ​​is 0.01, while the figures display forecast parameter values measured in thousands. Are these results derived from the same units of measurement? Has feature normalization been conducted, a common practice in machine learning tasks? If not, what is the rationale behind this omission?

7. An analysis of feature importance in training the model is lacking, making it difficult to determine which parameters were most influential in the development of the constructed neural network model;

8. The figures do not clarify whether the values presented pertain to the test or training sample.

Decoration:

1. There is an absence of references to the formulas;

2. Figures 1 and 2 lack appropriate citations;

3. The axes of the graphs are not labeled.

---

### Official Review · Reviewer_7Kqy · 2025-02-25
**Domain-specific multi-variate time series forecasting paper with weak experimental setup**

**Rating:** 6
**Confidence:** 3

**Review:**

The paper is devoted to the case study of time series forecasting from the field of energy industry. It focuses on GP-based methods.

The positive side of the paper is a solid description of the mathematical background. It is also good that open data is used.

For Figure 1, it is not quite clear to me why the predicted data (blue) are so close to the predicted (red) and at the same time the interval is quite wide. There are no references in the text for Figure 1, nor is there a detailed description of this issue.

Also, it is not clear to me why the comparison of the proposed models with other ML and statistical models for time series forecasting is avoided. It seems to be an important part of the experimental design to confirm the effectiveness of the proposed approach. Also, a simple baseline should be added to the comparison as a reference point.

It would also be useful to provide source code for experiments to increase reproducibility. At the double-blind review stage, https://anonymous.4open.science/ can be used.

---

### Official Review · Reviewer_Mtqg · 2025-02-26
**Review of "Probabilistic sparse variational model based on Gaussian processes for energy parameter forecasting" (updating)**

**Rating:** 6
**Confidence:** 2

**Review:**

This paper discusses the application of sparse variational Gaussian process (SVGP) models to predict energy parameters. The authors describe in detail the mathematical apparatus of Gaussian Processes, derive formulas for variational inference, and present experimental results with two data sets.

The paper has a number of significant shortcomings:
- The method for time series forecasting based on predictions is unclear.
- The modeling results are questionable, since confidence intervals do not increase from the beginning to the end of the forecast horizon.
- There are no specific formulas for calculating the forecasts, and there is no link to the program code for reproducing the results.
- There are no labels on the figures.

Update: Updated the rating in response to the authors' comments.

---

### Official Review · Reviewer_Ma9b · 2025-02-27
**The article presents a study focused on the application of a probabilistic sparse variational model based on Gaussian Processes (SVGP) for forecasting energy parameters. The work is highly relevant, as forecasting in the energy sector is a critical task, especially given the increasing instability and variability of energy consumption and generation. The authors propose the use of SVGP to account for uncertainty and improve the accuracy of forecasts, which represents a significant contribution to the field of energy forecasting.**

**Rating:** 8
**Confidence:** 5

**Review:**

1. General Assessment of the Article
The article presents a study focused on the application of a probabilistic sparse variational model based on Gaussian Processes (SVGP) for forecasting energy parameters. The work is highly relevant, as forecasting in the energy sector is a critical task, especially given the increasing instability and variability of energy consumption and generation. The authors propose the use of SVGP to account for uncertainty and improve the accuracy of forecasts, which represents a significant contribution to the field of energy forecasting.

2. Strengths of the Article
Relevance of the Topic: Forecasting energy parameters, particularly in the context of renewable energy, is an important task. The authors emphasize the need to account for uncertainty, making their approach particularly valuable.
Theoretical Foundation: The article provides a detailed description of the mathematical model, including the use of Gaussian Processes, variational inference, and inducing points. This makes the work scientifically sound and reproducible.
Practical Significance: The authors demonstrate the application of the model to real-world data, including forecasting wind energy generation and energy consumption. The results show high forecasting accuracy, as confirmed by metrics such as MAE, RMSE, and CRPS.
Use of Modern Methods: The application of the Matérn kernel and neural networks to enhance the predictive capability of the model demonstrates the use of contemporary approaches in machine learning.

3. Recommendations for Improvement
Expanding the Dataset: Including data from other regions or types of energy generation (e.g., solar) could improve the model's generalizability.
Deeper Analysis of Limitations: The authors should discuss the limitations of the model in more detail, including computational complexity and sensitivity to hyperparameters.
Comparison with Other Methods: Adding a comparison with other forecasting methods (e.g., LSTM, GRU, or other probabilistic models) would help better evaluate the advantages of the proposed approach.
Improved Visualization: The graphs of forecasts and uncertainty could be more detailed to better reflect the results.

4. Conclusion
The article represents a high-quality study that makes a significant contribution to the field of energy parameter forecasting. The use of SVGP to account for uncertainty and improve forecasting accuracy is a promising direction. To enhance the scientific value of the work, it is recommended to expand the dataset, conduct a deeper analysis of limitations, and compare the proposed model with other methods.

---

### Comment · Reviewer_9aHa · 2025-02-28
**Applying SVG for energy forecasting, a good mathematical part and not very clear experiments.**

Strengths of the work:
consideration of an important topic of energy parameters using extensive real data compiled specifically for the task, the mathematical part is well developed and described, in addition, the purpose of the work is clearly formulated and problematic parts in this area are noted.

Weak points of work:
the experimental part is not enough to draw full-fledged conclusions about uncertainty analysis and other things, several problem areas have been identified, but it is unclear how the results of the work respond to them, the drawings of the results are designed without axes, which complicates their interpretation, in addition, it would be good to add the code and dataset to the public domain, for example on github so that the results can be reproduced and unique training dataset can be used.

---

### Decision · Program_Chairs · 2025-03-08

**Decision:**

Accept (Oral)

**Comment:**

Your article has been accepted and you can give a talk on the article. All articles will be sorted by rating and within the available conference places one author from each article will be invited. If there are not enough places, then you will either have the opportunity to speak remotely or come at your own expense!